# Involvement of the Cerebellar Peduncles in *FMR1* Premutation Carriers: A Pictorial Review of Their Anatomy, Imaging, and Pathology

**DOI:** 10.3390/ijms26094402

**Published:** 2025-05-06

**Authors:** Irene Paracuellos-Ayala, Giovanni Caruana, Macarena Maria Reyes Ortega, Randi J. Hagerman, Jun Yi Wang, Laia Rodriguez-Revenga, Andrea Elias-Mas

**Affiliations:** 1Radiology Department, Hospital Universitari Mútua Terrassa (HUMT), Terrassa 08221, Spain; iparacuellos@mutuaterrassa.cat (I.P.-A.); gcaruana@mutuaterrassa.cat (G.C.); mreyes@mutuaterrassa.es (M.M.R.O.); aelias@mutuaterrassa.cat (A.E.-M.); 2Medical Investigation of Neurodevelopmental Disorders (MIND) Institute, University of California Davis, Sacramento, CA 95817, USA; rjhagerman@ucdavis.edu; 3Department of Pediatrics, University of California Davis Medical Center, Sacramento, CA 95817, USA; 4Center for Mind and Brain, University of California Davis, Davis, CA 95618, USA; jyiwang@ucdavis.edu; 5Biochemistry and Molecular Genetics Department, Hospital Clinic of Barcelona, 08036 Barcelona, Spain; 6CIBER of Rare Diseases (CIBERER), Instituto de Salud Carlos III, 08036 Barcelona, Spain; 7Fundació de Recerca Clínic Barcelona-Institut d’Investigacions Biomèdiques August Pi i Sunyer (IDIBAPS), 08036 Barcelona, Spain; 8Genetics Doctorate Program, Universitat de Barcelona (UB), 08036 Barcelona, Spain; 9Institute for Research and Innovation Parc Taulí (I3PT), 08208 Sabadell, Spain

**Keywords:** cerebellar peduncles, *FMR1* premutation, FXTAS, white matter

## Abstract

The cerebellar peduncles (CPs) contain essential pathways connecting the cerebellum and other regions of the central nervous system, yet their role is often overlooked in daily medical practice. Individuals with the *FMR1* premutation are at risk of developing fragile X-associated tremor/ataxia syndrome (FXTAS), a late-onset neurodegenerative disorder. The major clinical and radiological signs of FXTAS are cerebellar gait ataxia, intention tremor, and T2-weighted MRI hyperintensity of the middle cerebellar peduncle (MCP sign). Over the years, metabolic and structural abnormalities have also been described in the CPs of *FMR1* premutation carriers, with some being associated with CGG repeat length and *FMR1* mRNA levels. Evidence seems to associate the clinical disfunction observed in FXTAS with MCP abnormalities. However, other tracts within the different CPs may also contribute to the symptoms observed in FXTAS. By integrating imaging and pathological data, this review looks to enhance the understanding of the functional anatomy of the CPs and their involvement in different pathological entities, with special interest in premutation carriers and FXTAS. This review, therefore, aims to provide accessible knowledge on the subject of the CPs and their functional anatomy through detailed diagrams, offering a clearer understanding of their role in *FMR1* premutation.

## 1. Introduction

The cerebellar peduncles (CPs) are three paired white matter tracts that connect the brainstem and cerebellum, enabling the exchange of information essential for proper cerebellar function. Abnormalities in the CPs have been linked to numerous neurological disorders such as Multiple System Atrophy (MSA), Joubert Syndrome, spinocerebellar ataxias, or diffuse axonal injury [1,2,3,4,5,6], and the middle cerebellar peduncle (MCP) is the most extensively studied and reviewed [7,8,9,10].

The radiological delineation of the CPs can be challenging due to the presence of converging fiber tracts. The anatomy and connectivity of the tracts traversing the CPs are complex, yet their involvement in various pathological processes is critical for advancing our understanding of disease pathophysiology and clinical manifestations. Although the neuroanatomical pathways of the CPs are well described in classical textbooks [11,12], and numerous tractography studies have mapped their anatomical trajectories [4,13], along with several reviews focusing on pathology affecting the MCP [7,14], there remains a paucity of the literature schematizing the radiological–functional implications of abnormalities in the superior (SCPs), MCPs, and inferior cerebellar peduncles (ICPs).

*FMR1* premutation carriers (55–200 CGG repeats) are at risk for developing fragile X-associated tremor/ataxia syndrome (FXTAS), a late-onset neurodegenerative disorder, and the salient central nervous system (CNS) diagnostic feature is white matter hyperintensity in the MCP in males [15]. Clinically, FXTAS is characterized by progressive intention tremor, gait ataxia, and parkinsonian features, including bradykinesia and rigidity. Additional motor symptoms include dysarthria, dystonia, and peripheral neuropathy. Cognitive impairments, particularly deficits in executive function, working memory, and processing speed, are common and may progress to dementia. Psychiatric manifestations such as anxiety, depression, and mood dysregulation further contribute to disease burden. Autonomic dysfunction, including bladder and bowel disturbances and erectile dysfunction in males, is also frequently reported [16,17]. The severity and progression of these symptoms vary among individuals, but FXTAS ultimately leads to significant functional decline.

At the molecular level, FXTAS is caused by a toxic gain-of-function mechanism associated with elevated *FMR1* mRNA levels, leading to RNA toxicity rather than FMRP deficiency, which underlies fragile X syndrome [18]. The expanded CGG repeats in the *FMR1* gene result in excessive mRNA production, which forms intranuclear RNA foci that sequester RNA-binding proteins, such as heterogeneous nuclear ribonucleoproteins (hnRNPs) and Sam68, disrupting their normal cellular functions [19]. This sequestration impairs alternative splicing, protein translation, and RNA metabolism, contributing to widespread neuronal dysfunction [20,21]. Additionally, the *FMR1* mRNA with expanded CGG repeats is aberrantly translated via repeat-associated non-AUG (RAN) translation, producing toxic polyglycine-containing FMRpolyG proteins that aggregate in neuronal and glial nuclei, further exacerbating cellular toxicity [22,23].

Neuropathologically, FXTAS is characterized by eosinophilic intranuclear inclusions in neurons and astrocytes throughout the brain, particularly in the cerebellum, brainstem, and cerebral cortex [24,25]. These inclusions contain ubiquitin, heat shock proteins, and other misfolded proteins, indicating impaired proteostasis [26,27]. Furthermore, widespread white matter degeneration, axonal loss, and Purkinje cell dropout contribute to the progressive neurodegeneration observed in FXTAS. Mitochondrial dysfunction, oxidative stress, and neuroinflammation have also been implicated in disease pathology, highlighting the complex molecular mechanisms underlying FXTAS progression [28].

The severity and onset of FXTAS seems to correlate with the length of the CGG repeat expansion, with larger repeat sizes within the premutation range (typically 80–200 repeats) associated with an increased risk of developing FXTAS and more severe symptoms [29]. Individuals with higher CGG repeat lengths exhibit greater *FMR1* mRNA overexpression, leading to more extensive sequestration of RNA-binding proteins. This suggests that the cumulative burden of RNA toxicity and protein misfolding contributes to disease progression. However, efforts have been made to correlate clinical outcomes to CGG repeat length, and the relationship between the CGG repeat expansion and disease penetrance is complex, with other genetic and environmental factors influencing individual susceptibility and symptom variability [30,31].

Radiological findings in FXTAS are crucial for diagnosis and disease characterization, with the MCP sign serving as a key diagnostic marker. This hallmark feature appears as bilateral T2 hyperintensities in the MCPs on MRI, reflecting underlying white matter pathology and axonal degeneration. The MCP sign is present in a significant proportion of affected individuals and is considered a major radiological criterion for FXTAS diagnosis [15]. Beyond the MCP sign, widespread white matter abnormalities are commonly observed, particularly in the periventricular and subcortical regions. Hyperintensities on T2-weighted and FLAIR imaging extend into the cerebral white matter, splenium of the corpus callosum, and pons [32], indicating progressive demyelination and neurodegeneration. Generalized brain atrophy, including cerebellar and cerebral cortical thinning [15], is frequently reported and correlates with disease severity and cognitive decline. Enlargement of the lateral and third ventricles, a sign of brain volume loss or dysfunctional cerebrospinal fluid circulation, is also a common finding [33,34]. Advanced neuroimaging techniques, such as diffusion tensor imaging (DTI) and magnetic resonance spectroscopy (MRS), further reveal microstructural abnormalities in the MCPs of *FMR1* premutation carriers [35,36,37,38]. The clinical symptoms and radiological findings in *FMR1* premutation carriers suggest dysfunction of the cerebello-basal ganglia-thalamo-cortical network [33,39,40]. Notably, the MCP sign has been correlated with motor and cognitive impairment [41], which is consistent with the MCP’s role in transmitting pontocerebellar projections involved in motor planning, cognition, and language [7,11,42].

A deeper understanding of CP anatomy and function is essential for contextualizing the motor and cognitive impairments observed in *FMR1* premutation carriers and other neurological conditions. Each CP carries distinct types of information critical to cerebellar function. While the MCP primarily transmits cortical afferent fibers to the cerebellum, the SCP serves as the main efferent pathway, carrying signals from the dentate nuclei to the red nuclei, tectum, and thalamus [43,44]. In contrast, the ICP conveys afferent proprioceptive and somatosensory information to the cerebellum as well as afferent and efferent fibers to and from the reticular system [45].

Despite the evidence of pathological and clinical involvement of the CPs in FXTAS, with most of the efforts carried out on the MCP, there remains a lack of comprehensive diagrams that effectively integrate anatomical and functional insights with pathological findings. Therefore, this review aims to bridge this gap by providing a clear and accessible overview of CP functional anatomy and their involvement in neurological disorders, with a particular focus on the insights provided by *FMR1* premutation carriers and FXTAS.

## 2. Anatomy

The CPs connect the cerebellum to the brainstem, facilitating the passage of complex afferent and efferent tracts. These tracts exchange information with the thalamus, mesencephalic, pontine, and bulbar nuclei, ensuring proper cerebellar function. There are six CPs in total, three on each side: the SCP, MCP, and ICP (Figure 1).

The SCPs (Figure 2) are the primary efferent pathways from the cerebellum. Originating at the hilus of the dentate nuclei, they ascend to the midbrain, forming the posterolateral walls of the fourth ventricle. The SCPs enter the midbrain caudally to the inferior colliculus and trochlear nerves, decussate at the level of the inferior colliculus, and project to the midbrain nuclei and thalamus. Their blood supply comes from the superior cerebellar artery [12].

The MCPs (Figure 2) are the main afferent pathways to the cerebellum. Originating in the lateral pons and positioned laterally to the SCPs and ICPs, they form a wide, robust bundle that primarily projects to various regions of the cerebellar cortex. Their blood supply comes mainly from the superior cerebellar artery, with some contribution from the anterior inferior cerebellar artery [12].

The ICPs (Figure 2) carry both afferent and efferent information of the cerebellum, forming the inferior and lateral walls of the fourth ventricle. They consist of two components: the restiform body and the juxtarestiform body. The restiform body, purely afferent, ascends in the dorsolateral medulla, lateral to the vestibular nuclei and medial to the MCP. The juxtarestiform body, medial to the restiform body, carries both afferent and efferent fibers from the cerebellum to the vestibular nuclei in the medulla [11,12].

## 3. Fiber Tracts and Clinical Relevance

### 3.1. Superior Cerebellar Peduncle

As previously mentioned, the SCP is the major efferent pathway from the cerebellum (Figure 3). It carries tracts from the dentate and interpositus nuclei (globose and emboliform nuclei) that will mainly end in the contralateral red nuclei and thalamus [43,44]. Some of these tracts are the dentato-rubro-thalamic and the interposito-rubral tracts and will participate in modifying and coordinating motor skills and muscle activity from the same side of the body [42,46]. Some fibers from the interpositus and dentate nuclei exit the SCP and descend ipsilaterally through the pontomedullary tegmentum to reach the contralateral inferior olivary nuclei. This forms partly the cerebello-olivary pathway [11,12,47,48,49], a neural circuit crucial for refining motor activities and learning new motor skills through continuous feedback and error correction [42].

### 3.2. Middle Cerebellar Peduncle

Most of the SCP’s afferent tracts convey proprioceptive information from the trunk and lower limbs (anterior spinocerebellar tract), from the upper limbs (rostral spinocerebellar tract), and from the face (trigeminocerebellar tract). Additionally, the tectocerebellar tract conveys auditory, visual, and eye movement information, enhancing the accuracy and timing of movements based on these sensory inputs [42]. Other less representative tracts include the coeruleo-cerebellar tract, which is involved in stress-related noradrenergic modulation [50].

Damage to the SCP correlates with impaired motor functions in the limbs and walking ability [9,51,52], associating ataxia, tremor, and dysmetria [53,54,55,56,57,58]. Furthermore, the dentato-rubro-thalamic tract, implicated in tremor pathophysiology, has been effectively targeted by deep brain stimulation to reduce tremor [59,60]. Nystagmus, caused by damage to the tectocerebellar tract [61], and hypotonia may also be present when the SCP is impaired [62]. Interestingly, damage to the SCPs’ fibers has been linked to cerebellar mutism, a condition characterized by reduced or completely absent speech, typically occurring within the first week after surgery for a posterior fossa tumor [62,63,64]. The role of the SCPs’ fiber tracts in cognitive functions is not well understood, but a correlation has been reported between cognitive testing and altered fractional anisotropy (FA) of SCPs in patients with multiple sclerosis, schizophrenia, and preterm infants, three years after a low-grade intraventricular hemorrhage [65,66,67,68]. Furthermore, recent research has shown a positive association between procedural learning and microstructural organization of the SCP in healthy adults and children [69,70].

The MCP serves as the primary afferent pathway to the cerebellum, containing minimal efferent tracts (Figure 4). Notable components of the MCPs are the pontocerebellar projections, which transmit information from the contralateral cerebral cortex, via the pontine nuclei, to the cerebellum. These projections contribute to motor planning, cognition, and language functions. Other afferent tracts include those associated with the raphe nuclei and reticular nuclei, which play a role in the serotoninergic and noradrenergic modulation of various cerebellar connections [7,11,42]. Damage to the MCP significantly correlates with balance alterations [71,72] upper limb incoordination [73], cognitive impairment [74], and, eventually, could explain dysarthria [75]. These correlations appear to result from disrupted cerebellar access to motor, cognitive, and limbic afferent information from the cerebral cortex via the MCP.

### 3.3. Inferior Cerebellar Peduncle

Similar to the SCP, the ICP (Figure 5) transmits proprioceptive information from the trunk and lower limbs (posterior spinocerebellar tract) as well as from the upper limbs (rostral spinocerebellar tract), neck, and head (cuneocerebellar tract). It also plays a crucial role in carrying information from the vestibular system through the vestibulocerebellar tract. Other afferents would include the reticulocerebellar and olive-cerebellar tracts that would transmit integrated somatosensorial information [11,12].

Damage to the ICP leads to postural imbalance with vertigo and nystagmus [76] as well as poor walking ability [52,77].

## 4. Imaging of the Cerebellar Peduncles

The CPs can be evaluated for pathology using MRI signal characteristics, morphological assessments (e.g., width measurements), and more advanced techniques such as diffusion, tractography, or spectroscopy.

MRI T2/FLAIR sequences: T2/FLAIR hyperintensities allow the visual assessment of macrostructural damage of the CPs. Hyperintensity, when present, indicates abnormality and may reflect demyelination, Wallerian degeneration, cytotoxic edema, vasogenic edema, etc. [78].Basic MRI DWI/ADC sequences: These sequences provide information about the water diffusion properties, aiding in characterization of the evolutionary phase of ischemic lesions and supporting differential diagnosis of space-occupying lesions [79].Diffusion MRI and tractography: Diffusion MRI is a relatively new MRI technique that allows for the study of white matter microarchitecture. Tractography is a 3D visualization technique to reconstruct white matter fiber tracts using data collected by diffusion MRI. It enables the visualization of cerebellar tract directionality and decussation and allows the calculation of FA, a key metric that quantifies the directional coherence of water diffusion within a voxel. FA ranges from 0 (isotropic, random diffusion) to 1 (anisotropic, organized tracts). However, FA does not solely reflect integrity—crossing fibers in a voxel can lower FA despite intact tracts, while high FA might reflect loss of one fiber tract, not improved health. Complementary DTI metrics help refine interpretation [80]. A high angular diffusion data acquisition scheme, such as High Angular Resolution Diffusion Imaging (HARDI), together with multicompartment orientation reconstruction methods, such as the multi-shell multi-tissue spherical deconvolution method [81], can resolve multiple intravoxel fiber orientations and are particularly useful in regions with crossing fibers, where traditional diffusion tensor models (assuming a single fiber population per voxel) fall short in capturing the underlying macrostructural tissue complexity [82,83,84].MR Spectroscopy: This technique analyzes brain metabolites, providing insight into neuronal integrity and cellular composition. The N-acetylaspartate/Creatine (NAA/Cr) ratio reflects neuronal health, with reductions indicating neurodegeneration, while the Choline/Creatine (Ch/Cr) ratio represents membrane turnover, aiding in the assessment of demyelination and tumor characterization [85].Peduncular width: Some studies have reported average values in healthy population measured in T1-weighted sequences. For individuals with a median age of 60.75, with standard deviation (SD) of 9.95 (*n* = 61), or older (*n* = 48), the SCP should be measured in the coronal plane, with normal values of 5.09 ± 0.82 mm (SD) or in the axial plane at the level of the inferior colliculus (2.2 ± 0.46 mm). The MCP should be measured in parasagittal slices (9.61 ± 1.1 mm) or in the axial plane at the level of the trigeminal nerve (13 ± 1.8 mm). The ICP should be measured in the axial plane at the level of the connection between the cerebellum and the medulla (5 ± 0.12 mm) [8,86].

## 5. Pathological Involvement of the Cerebellar Peduncles in FMR1 Premutation

Some studies have elucidated the role of CPs in *FMR1* premutation carriers, particularly focusing on structural and functional abnormalities. Key findings include the following:

Neuropathological examination: Autopsy studies of the MCPs in FXTAS revealed spongiosis, reflecting neurodegeneration [25].

Signal abnormalities in the MCP: The most recognized radiological feature for FXTAS is the MCP sign (symmetrical T2/FLAIR hyperintense signal in the MCP) (Figure 6). The MCP sign is seen in 58% of males and 13% of females with FXTAS [15,38]. Some publications reported this sign in asymptomatic *FMR1* premutation carriers [35,87], suggesting that it might appear in early preclinical stages of FXTAS.

Morphological features in the MCP: MCP width as well as midbrain and pons cross-sectional area has been shown to be reduced in patients with FXTAS compared to both premutation carriers without FXTAS and controls. Furthermore, decreased MCP width has been suggested as a potential biomarker to identify carriers at risk to develop FXTAS [8].Structural abnormalities in the SCP, MCP, and ICP and its correlation with molecular data: Significant reductions in FA and elevation of diffusivity have been described in the MCP and SCP of *FMR1* premutation carriers with FXTAS [36,37,88]. The reported significant elevation of diffusivity measures in *FMR1* premutation carriers without FXTAS [89,90] suggests preclinical change in white matter microarchitecture that warrants confirmation in longitudinal studies. Inverted U-shaped correlation between diffusivity measures and CGG repeat length was also demonstrated [36] as well as a negative dose effect of CGG repeat length and *FMR1* mRNA on the connectivity strength of SCPs [37]. Negative correlation between the circulating *FMR1* mRNA level and mean diffusivity in the MCP was also demonstrated in female premutation carriers without FXTAS. Additionally, decreased mean diffusivity in the MCP and ICP showed significant correlation with higher methylation levels in the *FMR1* gene [40]. Currently, this is the only study that revealed *FMR1* molecular correlation in the ICP.Metabolic abnormalities in the MCP: Significant decreased levels of metabolites NAA/Cr and Ch/Cr in the MCP of *FMR1* premutation carriers have been described, plausibly representing axonal loss and demyelination [35].Clinical correlation: The MCP sign and microstructural white matter abnormalities observed in the SCP and MCP as determined by MRS and DTI studies have shown significant correlation with executive dysfunction, slow processing speed, dexterity, and cognition dysfunction in *FMR1* premutation carriers [35,37].

## 6. Other Pathological Entities Affecting the Cerebellar Peduncles

Beyond FXTAS, several other conditions can affect the CPs (Table 1) through inherited or acquired mechanisms, including vascular, inflammatory, metabolic, and neurodegenerative processes. Joubert Syndrome features SCP elongation and the ‘molar tooth sign’ (Figure 7) [91], while Progressive Supranuclear Palsy and MSA show SCP/MCP atrophy and pontine signal changes (e.g., ‘hot cross bun sign’) [3,92,93,94,95]. Spinocerebellar ataxias impair all three CPs [2,96,97], and conditions like cerebral autosomal recessive arteriopathy with subcortical infarcts and leukoencephalopathy (CARASIL) and chronic lymphocytic inflammation with pontine perivascular enhancement responsive to steroids (CLIPPERS) exhibit distinct MCP/pons hyperintensities [10,98]. Diffuse axonal injury frequently targets the CPs [99], and emerging studies link ICP/MCP changes to ADHD, schizophrenia, and language disorders [5,100,101,102,103,104,105,106]. Ischemic insults can also rarely isolate the CPs (Figure 8).

## 7. Conclusions

The abnormalities observed in the CPs of *FMR1* premutation carriers primarily converge on white matter degeneration. Evidence suggests that this process likely begins with microscopic changes in the CPs, potentially detectable via DTI and MRS, followed by reduced peduncular volume or width, and culminating in altered T2-FLAIR signals. However, the temporal sequence of these findings needs to be confirmed with imaging longitudinal studies.

Neuropathological studies in FXTAS have demonstrated spongiosis in the MCPs and deep cerebellar white matter [25]. Imaging studies further support a correlation between the presence of the MCP sign and CGG repeat length [41], as well as negative associations between CGG repeat length and FMR1 mRNA levels with the structural connectivity of the SCPs in premutation carriers [37]. Additionally, the MCP sign has been associated with impaired motor and executive functioning [37]. These findings suggest that CGG repeat length and elevated FMR1 mRNA levels may contribute to RNA toxicity, potentially disrupting axonal transport and leading to white matter pathology in the CPs and subsequent motor impairment. However, further studies are needed to clarify the specific molecular pathways underlying pathophysiological changes in the brain.

The SCP, MCP, and ICP are closely interconnected, with the SCP and ICP narrower than the MCP. As a result, the larger size and greater white matter content of the MCP may explain the more consistent imaging abnormalities observed in the MCP, such as pronounced hyperintense signals, which could obscure subtle white matter alterations in the SCP and ICP. The ICP, in particular, remains understudied, and its involvement may be underestimated due to these factors. Additionally, the imprecise delineation of CP anatomical boundaries in conventional radiological imaging poses challenges to assessment accuracy, potentially leading to inconsistent research findings. Advanced imaging techniques such as high-resolution diffusion MRI or quantitative MRI, are thus needed to perform future studies on the SCP and ICP.

Understanding the distinct neuroanatomical pathways within each CP enhances insight into the symptomatology of FXTAS. Damage to the pontocerebellar fibers within the MCP likely disrupts cortical input to the cerebellum, impairing precise motor function execution. Similarly, injury to proprioceptive and somatosensory tracts entering the cerebellum via the SCP and ICP may contribute to ataxia and movement incoordination [51,52,54,77]. Alterations in the SCP and MCP are well documented in *FMR1* premutation carriers [15,36,37,89,90], and emerging data suggest potential involvement of the ICP as well [40]. However, further research is needed to confirm these observations and clarify the full scope of CP involvement.

This pictorial review strengthens these findings by providing figures of the CPs’ functional anatomy in a simple, yet comprehensive manner, hoping to facilitate the scientific community’s study and interpretation of findings involving the CPs. By bridging neuroanatomic, radiologic, clinical, and molecular data, this review not only resumes current evidence but also highlights areas of uncertainty, such as the extent of SCP/ICP involvement in *FMR1* premutation carriers. Future studies employing longitudinal designs and advanced imaging techniques will be critical to confirm these observations, elucidate progression patterns, and explore therapeutic strategies.

## Figures and Tables

**Figure 1 ijms-26-04402-f001:**
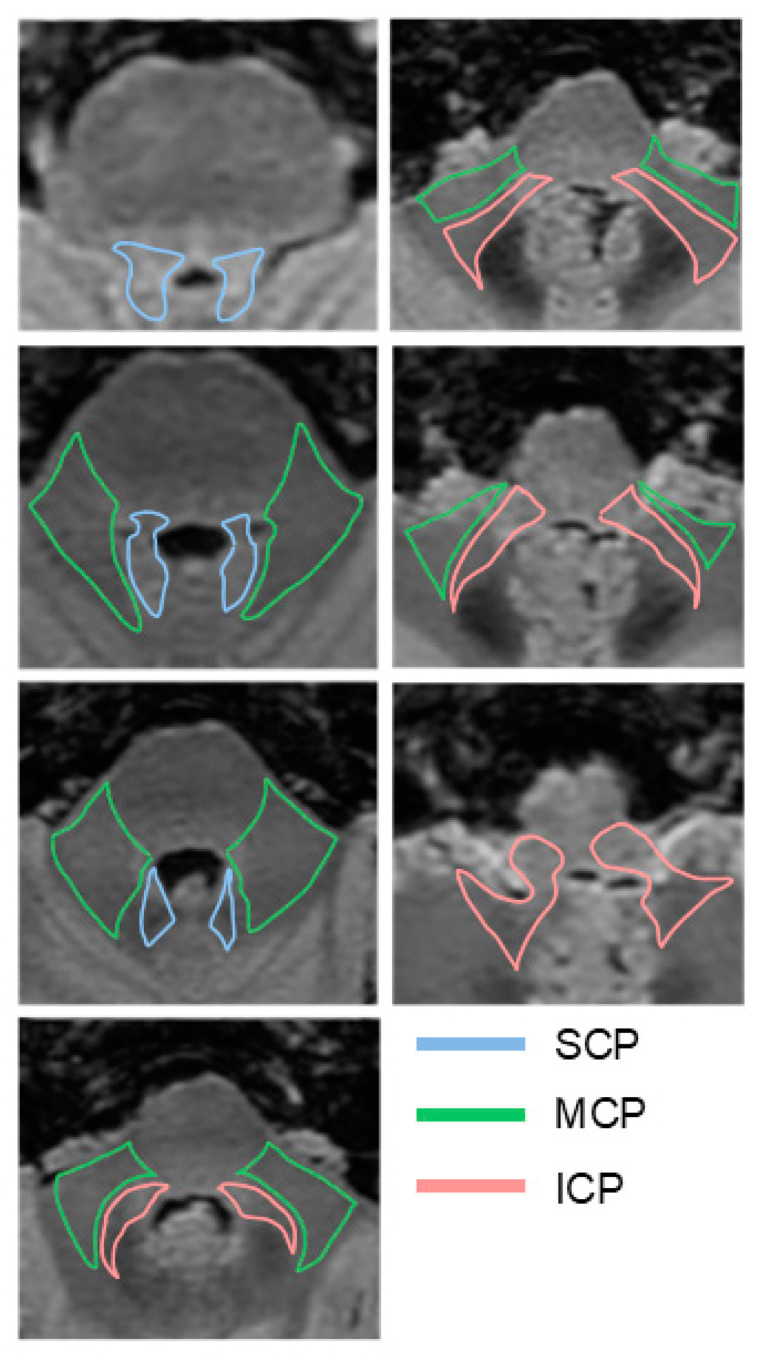
Simplified schematic showing MRI T2 FLAIR axial images of the brainstem with delineation of the SCP, MCP, and ICP trajectories.

**Figure 2 ijms-26-04402-f002:**
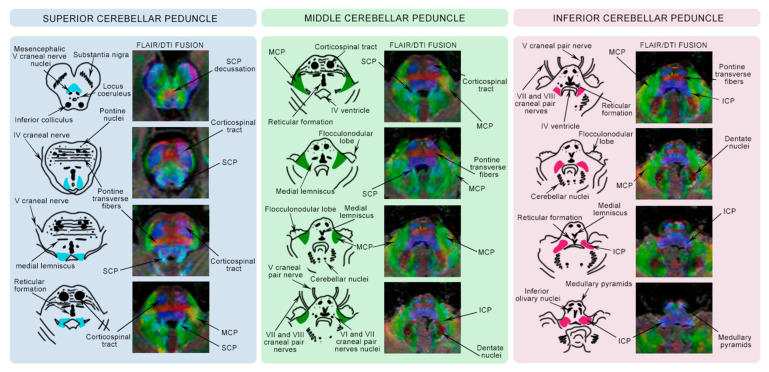
MRI images of the brainstem using combined T2 FLAIR sequences and DTI at different anatomical levels for each CP. On the left of each colored column, a pictorial representation of the corresponding CP is shown. The DTI color-coded map uses the standard color-coding convention: red represents transverse fibers, blue represents craniocaudal fibers, and green represents anteroposterior fibers.

**Figure 3 ijms-26-04402-f003:**
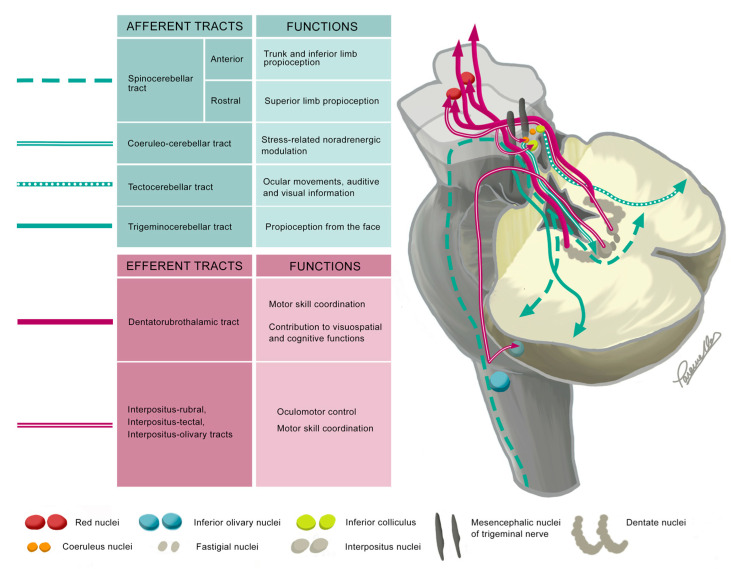
The diagram illustrates the most relevant tracts within the SCPs and their functions, using a representation of the brainstem and the lower half of the cerebellum. The upper part of the cerebellum is cut off at the level of the SCP and MCP. To enhance anatomical clarity, some tracts are shown unilaterally. Afferent tracts are colored green, while efferent fibers are colored magenta.

**Figure 4 ijms-26-04402-f004:**
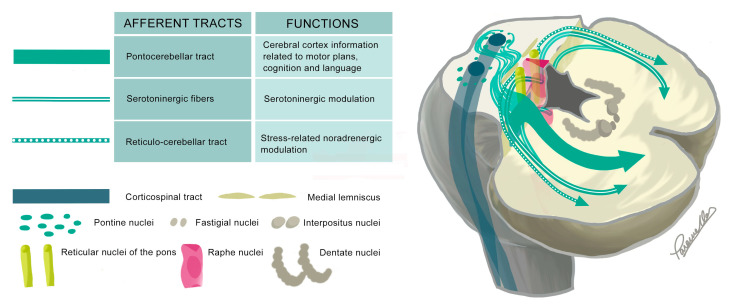
The diagram illustrates the most relevant tracts within the MCPs and their functions using a representation of the brainstem at the level of the pons and the lower half of the cerebellum. The nuclei are shown bilaterally, while the tracts are depicted unilaterally. In the diagram, afferent tracts are colored green. Efferent tracts are scarce and thus not represented.

**Figure 5 ijms-26-04402-f005:**
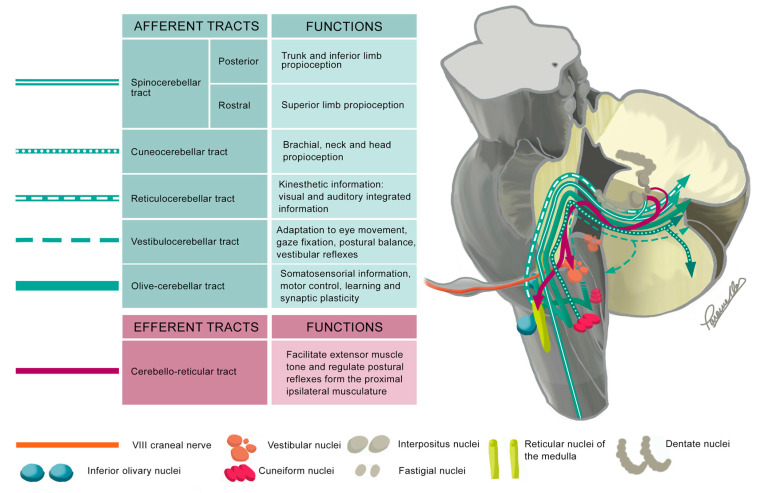
The diagram illustrates the most relevant tracts within the ICPs using a representation of the brainstem and a portion of the cerebellum, which is sectioned to better visualize the neuroanatomy of the CPs. The nuclei are shown bilaterally, while the tracts are depicted unilaterally, specifically for the left ICP. In the diagram, afferent tracts are colored green, and efferent fibers are colored magenta.

**Figure 6 ijms-26-04402-f006:**
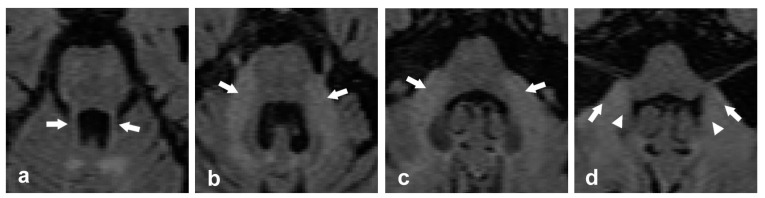
Neuroimaging axial T2 FLAIR findings in a 73-year-old male with FXTAS (stage 5) who exhibits intentional and resting tremors, parkinsonism, cerebellar ataxia, and deficits in attention, concentration, and working memory. Axial T2 FLAIR images show (**a**) the SCPs (arrows), (**b**) hyperintense signal at the MCPs (arrows), commonly known as the MCP sign, and (**c**,**d**) hyperintensities at both the MCPs (arrows) and ICPs (head arrows).

**Figure 7 ijms-26-04402-f007:**
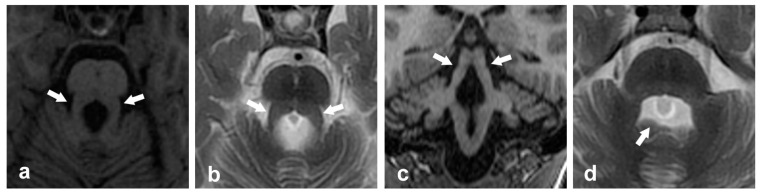
Neuroimaging findings in an 8-year-old child with Joubert Syndrome who experienced psychomotor developmental delay the first months of life and suffers from postural instability and gait ataxia are as follows: (**a**) axial T1-weighted image and (**b**) axial T2-weighted image show the thickened, elongated, and horizontally orientated SCP (arrows) at the level of the ponto-mesencephalic junction (molar tooth sign), (**c**) coronal T1-weighted image displays thickened SCP (arrows), and (**d**) axial T2-weighted image reveals a bat wing-shaped 4th ventricle (arrow).

**Figure 8 ijms-26-04402-f008:**
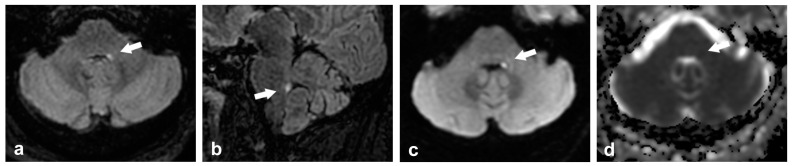
Neuroimaging findings in a 57-year-old patient with an acute small stroke in the left ICP. He presented to the emergency room with acute binocular diplopia, cephalic and gait instability, motion sickness, and vomiting. (**a**) Axial T2 FLAIR image that shows a small hyperintense foci in the left ICP (arrow), (**b**) sagittal T2 FLAIR image at the level of the hyperintense foci in the ICP (arrow), (**c**) axial DWI sequence, and (**d**) axial ADC sequence shows subtle restriction to diffusion (arrows).

**Table 1 ijms-26-04402-t001:** Table summarizing the main abnormalities of the CPs in different neurological disorders.

Neurological Disorder	Alterations	CNS Atrophy Pattern	References
**FXTAS**	Reduced FA in all CPs Reduced width of the MCP MCP sign	Generalized brain and cerebellar atrophy	[8,15,33,34,36,37,40]
**Joubert’s syndrome**	Absence of decussation of the SCP tracts Elongation, horizontalization, and increased width of the SCP, forming the molar tooth sign	Not applied (hypo-dysplasia of the cerebellar vermis)	[91]
**MSA (MSA-C variant)**	Reduced width of the SCP and MCP Reduced FA in the MCP Pontine cruciform hyperintensities (hot cross bun sign) MCP sign T2 hyperintensity of the ICP (ICP sign)	Brainstem Cerebellum	[92,94,95,107]
**Progressive supranuclear palsy**	Reduced FA and reduced width of the SCP	Midbrain	[3]
**Spinocerebellar ataxia**	Reduced FA in all CPs Reduced width of the MCP Hot cross bun sign MCP sign	Pons Cerebellum	[2,14,96,97,107]
**CARASIL**	Symmetrical T2/FLAIR hyperintense signal in the MCP connecting through the pons (arc sign)	Brain Brainstem Cerebellum	[10]
**CLIPPERS**	MRI punctate Pattern of patchy gadolinium enhancement ‘peppering’ the brainstem and MCP	Not characteristic at early stages	[98]
**Diffuse axonal injury**	Reduced FA in all CPs Subtle hyperintense small lesions on T2 weighted image and/or hypointense on T2*-weighted image ^1^ (microbleeds)	Not characteristic at early stages	[99]

^1^ T2*-weighted image is a gradient recalled echo (GRE) MRI sequence sensitive to T2* relaxation and susceptibility effects.

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
