# Peer review of "Involvement of the Cerebellar Peduncles in FMR1 Premutation Carriers: A Pictorial Review of Their Anatomy, Imaging, and Pathology"

_ijms, 2025, doi:10.3390/ijms26094402_

Round 1
Reviewer 1 Report
Comments and Suggestions for Authors
The authors present a comprehensive review of cerebellar peduncle (CP) involvement in FMR1 premutation carriers, with a strong focus on imaging findings. The manuscript is well-structured and offers a solid synthesis of MRI data. I recommend publication after the following revisions.
- On Page 9, peduncular width measurements include standard deviations but lack methodological details such as sample size, imaging protocols, and anatomical alignment. Please clarify how planes were selected (e.g., axial or coronal) to ensure reproducibility. Also, correct the typographical error at the end of Line 296 (missing closing parenthesis).
- The discussion highlights correlations between CGG repeat length, mRNA levels, and imaging findings, but lacks mechanistic insight. Including a paragraph on potential causal pathways (e.g., RNA foci disrupting axonal transport in CP tracts) would enhance the depth of interpretation.
- The manuscript notes clinical correlations (e.g., MCP abnormalities and cognitive deficits) but would benefit from expanded discussion of the underlying neuroanatomy, such as disrupted cortico-ponto-cerebellar circuits or dentato-rubro-thalamic tract damage in relation to tremor.
- Grouped citations in the Introduction (e.g., [20,25,28–31]) obscure the specific source of individual claims. To improve clarity, assign references to distinct findings (e.g., “[20] reports spongiosis; [25] links MCP hyperintensity to clinical staging”).
- Minor typographical errors remain, such as “CCG repeat lenght” (Page 2) and “Interpositus rubral tracts” (Page 5). A careful proofreading is recommended.
- Some abbreviations, such as “FA” (fractional anisotropy), are defined multiple times (e.g., on Page 9). Please ensure each abbreviation is defined only once at first mention and used consistently thereafter.
- The limitations of FA in regions with crossing fibers (e.g., SCP/ICP) are mentioned but deserve further emphasis. Consider discussing how advanced diffusion models such as HARDI may mitigate this issue.
- Normative values cited on Page 9 lack demographic context. Please clarify whether they derive from healthy controls, patient cohorts, or mixed samples, and include age range and sample size where available.
- Section 6 introduces other CP-related disorders (e.g., Joubert syndrome, MSA) but does not compare their imaging features to FXTAS. A comparative table summarizing key differences in T2 hyperintensity patterns and atrophy distribution would be helpful for differential diagnosis.
Author Response
The authors present a comprehensive review of cerebellar peduncle (CP) involvement in FMR1 premutation carriers, with a strong focus on imaging findings. The manuscript is well-structured and offers a solid synthesis of MRI data. I recommend publication after the following revisions.
1. On Page 9, peduncular width measurements include standard deviations but lack methodological details such as sample size, imaging protocols, and anatomical alignment. Please clarify how planes were selected (e.g., axial or coronal) to ensure reproducibility. Also, correct the typographical error at the end of Line 296 (missing closing parenthesis).
Response: Thank you very much for your insightful feedback. I have added details regarding the sample size, imaging protocol (MRI T1-weighted sequences), and anatomical alignment to enhance reproducibility. Specifically, I clarified the plane levels for each cerebellar peduncle (CP) measurement, ensuring that the coronal, axial, or parasagittal planes are clearly defined as appropriate. Additionally, I have corrected the typographical error on Line 319 (previously 296) by adding the missing closing parenthesis.
2. The discussion highlights correlations between CGG repeat length, mRNA levels, and imaging findings, but lacks mechanistic insight. Including a paragraph on potential causal pathways (e.g., RNA foci disrupting axonal transport in CP tracts) would enhance the depth of interpretation.
Response: In accordance with your recommendation, we have added a paragraph to the conclusion section discussing potential causal mechanisms by which molecular abnormalities in FMR1 premutation carriers may contribute to neurodegeneration in the cerebellar peduncle tracts.
3. The manuscript notes clinical correlations (e.g., MCP abnormalities and cognitive deficits) but would benefit from expanded discussion of the underlying neuroanatomy, such as disrupted cortico-ponto-cerebellar circuits or dentato-rubro-thalamic tract damage in relation to tremor.
Response: Identifying publications that correlate isolated CP abnormalities with clinical outcomes has been challenging. Even more difficult has been finding studies on specific tracts to establish precise associations between tract damage and symptoms.
In response to your recommendation, I have revised this section to include a discussion on the role of the dentato-rubro-thalamic tract in tremor pathophysiology. Additionally, I have incorporated new information on the microstructural organization of the SCP and its positive association with procedural learning. Finally, I briefly discussed how MCP damage may contribute to balance alterations, incoordination, and cognitive impairment by disruption of cortico-ponto-cerebellar circuits.
4. Grouped citations in the Introduction (e.g., [20,25,28–31]) obscure the specific source of individual claims. To improve clarity, assign references to distinct findings (e.g., “[20] reports spongiosis; [25] links MCP hyperintensity to clinical staging”).
Response: We revised and organized citations to properly reference individual claims, as you indicated, trying, in some cases, to avoid detailed discussion that is going to be developed afterwards.
5. Minor typographical errors remain, such as “CCG repeat lenght” (Page 2) and “Interpositus rubral tracts” (Page 5). A careful proofreading is recommended.
Response: We have corrected "CGG repeat lenght" to "CGG repeat length and corrected the term “interpositus rubral tracts” to “interposito-rubral tracts”. We found some other minor typographical errors throughout the manuscript and in Figure 5, so we made the corrections accordingly. I also changed Figure 5 for a new version without tyographical errors and uploaded the new version in the resubmittion page. We,’ve proofread the manuscript as recommended to ensure no further errors.
6. Some abbreviations, such as “FA” (fractional anisotropy), are defined multiple times (e.g., on Page 9). Please ensure each abbreviation is defined only once at first mention and used consistently thereafter.
Response: We have revised the manuscript to ensure that all abbreviations, including "FA" (fractional anisotropy), are defined only once at their first mention and used consistently thereafter. We also removed the abbreviations in the section titles 4-6 and change it to the full term, assuring clarity for readers.
7. The limitations of FA in regions with crossing fibers (e.g., SCP/ICP) are mentioned but deserve further emphasis. Consider discussing how advanced diffusion models such as HARDI may mitigate this issue.
Response: Thank you for the suggestion. We have incorporated a brief explanation of how advanced diffusion models, such as High Angular Resolution Diffusion Imaging (HARDI), can mitigate these challenges by enhancing the accuracy of tractography in these complex regions.
8. Normative values cited on Page 9 lack demographic context. Please clarify whether they derive from healthy controls, patient cohorts, or mixed samples, and include age range and sample size where available.
Response: We have added the requested details on Page 9, as outlined in our response to point 1, ensuring that the normative values are now clearly attributed to healthy controls. We also made some changes in defining the standard deviation after each corresponding value.
9. Section 6 introduces other CP-related disorders (e.g., Joubert syndrome, MSA) but does not compare their imaging features to FXTAS. A comparative table summarizing key differences in T2 hyperintensity patterns and atrophy distribution would be helpful for differential diagnosis.
Response: Thank you for your valuable suggestion. Although the purpose of Section 6 was to provide a brief overview of cerebellar peduncle involvement in other disorders, without delving into extensive detail, we considered that adding a summarizing table would help comprehend the previous information. We hope that the table meets the idea that you had.
Reviewer 2 Report
Comments and Suggestions for Authors
Comments and suggestions for authors
The review article “Involvement of the Cerebellar Peduncles in FMR1 Premutationcarriers: A Pictorial Review of their Anatomy, Imaging and Pathology” is written in a very general form. It is easy to read and is more supported by the abbreviations provided at the end. Here are my suggestions and comments
The introduction provides an overview of the field and context for the topic. Authors should emphasize the importance of the topic they are reviewing, keeping the aim meaningful and clear with sufficient details. It should cover the main points in detail and must evaluate its sources. The authors have referenced many sources in the articles, but there are no logical arguments that support the topic.
The review article has 94 references, and a few cover recent research papers. The authors should add more recent research papers from the past 2 years and make a compelling discussion.
Furthermore, authors should elaborate on the review aims, adding more information following the last paragraph. The authors should provide references to the statement “ Although, efforts have been made to correlate clinical outcomes to CGG repeat lenght, the relationship between the CGG repeat expansion and disease
penetrance is complex, with other genetic and environmental factors influencing
individual susceptibility and symptom variability.” Also, based on the references, discuss the efforts that have been made to correlate clinical outcomes to CCG repeat length, the findings obtained, and how they are correlated and relevant to the topic you are reviewing.
There is a spelling error in line 90. Change “lenght” to “length”
I recommend enhancing the quality of the review article to make it more scientifically readable rather than just adding pictorial diagrams. Additionally, please cite the references at the end of the figure captions. I don't have any further suggestions or comments.
Thanks,
Best wishes
Author Response
Comment 1: The review article “Involvement of the Cerebellar Peduncles in FMR1 Premutation carriers: A Pictorial Review of their Anatomy, Imaging and Pathology” is written in a very general form. It is easy to read and is more supported by the abbreviations provided at the end. Here are my suggestions and comments
Response: Thank you for your thoughtful feedback and valuable suggestions. Below, we address each of your comments and outline the revisions made in response.
Comment 2: The introduction provides an overview of the field and context for the topic. Authors should emphasize the importance of the topic they are reviewing, keeping the aim meaningful and clear with sufficient details. It should cover the main points in detail and must evaluate its sources. The authors have referenced many sources in the articles, but there are no logical arguments that support the topic.
Response: To address this, we have revised the introduction to better emphasize the importance of the topic and provide a more detailed and logical evaluation of the sources. Specifically, we have added a second paragraph to better emphasize the significance and relevance of the topic addressed. Additionally, after presenting the current state of knowledge—particularly focusing on FXTAS—we have revised the final paragraph to clearly and explicitly state the objective of the study.
Comment 3: The review article has 94 references, and a few cover recent research papers. The authors should add more recent research papers from the past 2 years and make a compelling discussion.
Response: Regarding the references, we acknowledge your concern about the inclusion of more recent research. While the manuscript initially included 94 references, many of which were foundational to contextualizing the neuroanatomy and pathology of CPs in FXTAS, we recognize that only a few were from the past two years. In the new version f the manuscript, we have included 14 references from the past two years: 8 from 2024 and 6 from 2023, comprising recent research studies and reviews. We believe these additions enrich the discussion by highlighting broader clinical correlations of CP abnormalities, and we have integrated them into a more compelling discussion in the relevant sections.
Comment 4: Furthermore, authors should elaborate on the review aims, adding more information following the last paragraph. The authors should provide references to the statement “ Although, efforts have been made to correlate clinical outcomes to CGG repeat length, the relationship between the CGG repeat expansion and disease penetrance is complex, with other genetic and environmental factors influencing individual susceptibility and symptom variability.” Also, based on the references, discuss the efforts that have been made to correlate clinical outcomes to CCG repeat length, the findings obtained, and how they are correlated and relevant to the topic you are reviewing.
Response: In the revised manuscript, we have expanded the final paragraph of the introduction to clearly articulate the aims of the review. To support the statement regarding the complexity of the relationship between CGG repeat expansion and disease penetrance, we have added two relevant references that underscore the influence of additional genetic and environmental factors on individual susceptibility and symptom variability. Furthermore, we have added a new paragraph in the conclusion section that discusses current efforts to correlate clinical and molecular parameters—including CGG repeat length and FMR1 mRNA levels—with cerebellar peduncle pathology. This addition provides a more comprehensive synthesis of how these correlations contribute to our understanding of disease mechanisms in FXTAS and supports the relevance of the topic reviewed.
Comment 5: There is a spelling error in line 90. Change “lenght” to “length”
Response: We have corrected the spelling error on line 101 (previously line 90) (“lenght” to “length”) and conducted a thorough proofreading of the manuscript to ensure no other typographical errors remain. Additionally, we identified and corrected several minor typographical errors throughout the manuscript and in Figure 5. The revised figure has been updated in the manuscript and uploaded in the corresponding section of the resubmission page.
Comment 6: I recommend enhancing the quality of the review article to make it more scientifically readable rather than just adding pictorial diagrams. Additionally, please cite the references at the end of the figure captions. I don't have any further suggestions or comments.
Response: We appreciate your recommendation to enhance the scientific readability of the review beyond the inclusion of pictorial diagrams. With the revisions made, we believe the manuscript now strikes a better balance, offering a robust synthesis of evidence on the cerebellar peduncles and FXTAS that will be valuable for researchers seeking to deepen their understanding of this topic and guide future investigations. We view the pictorial diagrams not as the sole focus but as a complementary tool to visually illustrate the anatomical and radiological findings discussed, thereby enhancing the review’s utility.
Finally, we seek clarification regarding your suggestion to cite references at the end of the figure captions. As the author of these figures, I created them specifically for this review based on previous data and anatomical book (already referenced along the review). Are there any specific citation requirements for original figures in this context?
Thank you again for your constructive feedback, which has significantly improved the manuscript. We hope these revisions address your concerns and enhance the overall quality of the review.
Best regards,